

# *hdac4* mediates perichondral ossification and pharyngeal skeleton development in the zebrafish

April DeLaurier, Cynthia Lizzet Alvarez and Kali J Wiggins

Department of Biology and Geology, University of South Carolina—Aiken, Aiken, SC, United States of America

## ABSTRACT

**Background**. Histone deacetylases (HDACs) are epigenetic factors that function to repress gene transcription by removing acetyl groups from the N-terminal of histone lysines. Histone deacetylase 4 (HDAC4), a class IIa HDAC, has previously been shown to regulate the process of endochondral ossification in mice via repression of Myocyte enhancer factor 2c (MEF2C), a transcriptional activator of *Runx2*, which in turn promotes chondrocyte maturation and production of bone by osteoblasts.

**Methods & Materials**. In this study, we generated two zebrafish lines with mutations in *hdac4* using CRISPR/Cas9 and analyzed mutants for skeletal phenotypes and expression of genes known to be affected by Hdac4 expression.

**Results**. Lines have insertions causing a frameshift in a proximal exon of *hdac4* and a premature stop codon. Mutations are predicted to result in aberrant protein sequence and a truncated protein, eliminating the Mef2c binding domain and Hdac domain. Zygotic mutants from two separate lines show a significant increase in ossification of pharyngeal ceratohyal cartilages at 7 days post fertilization (dpf) ($p < 0.01$, $p < 0.001$). At 4 dpf, mutant larvae have a significant increase of expression of *runx2a* and *runx2b* in the ceratohyal cartilage ($p < 0.05$ and $p < 0.01$, respectively). A subset of maternal-zygotic (mz) mutant and heterozygote larvae (40%) have dramatically increased ossification at 7 dpf compared to zygotic mutants, including formation of a premature anguloarticular bone and mineralization of the first and second cerato-branchial cartilages and symplectic cartilages, which normally does not occur until fish are approximately 10 or 12 dpf. Some maternal-zygotic mutants and heterozygotes show loss of pharyngeal first arch elements (25.9% and 10.2%, respectively) and neurocranium defects (30.8% and 15.2%, respectively). Analysis of RNA-seq mRNA transcript levels and *in situ* hybridizations from zygotic stages to 75–90% epiboly indicates that *hdac4* is highly expressed in early embryos, but diminishes by late epiboly, becoming expressed again in larval stages.

**Discussion**. Loss of function of *hdac4* in zebrafish is associated with increased expression of *runx2a* and *runx2b* targets indicating that a role for *hdac4* in zebrafish is to repress activation of ossification of cartilage. These findings are consistent with observations of precocious cartilage ossification in *Hdac4* mutant mice, demonstrating that the function of *Hdac4* in skeletal development is conserved among vertebrates. Expression of *hdac4* mRNA in embryos younger than 256–512 cells indicates that there is a maternal contribution of *hdac4* to the early embryo. The increase in ossification and profound loss of first pharyngeal arch elements and anterior neurocranium in a subset of maternal-zygotic mutant and heterozygote larvae suggests that maternal

Corresponding author
April DeLaurier, aprild@usca.edu

*hdac4* functions in cartilage ossification and development of cranial neural crest-derived structures.

## INTRODUCTION

The majority of the vertebrate skeleton including the axial, limb, and pharyngeal elements form as cartilaginous elements that grow rapidly during early development through the proliferation of matrix-secreting chondrocytes (*Karsenty & Wagner, 2002*). At specific stages of development, chondrocytes cease their rapid proliferation and matrix secretion, become hypertrophic, and signal to nearby cells to commence endochondral ossification (*Karsenty & Wagner, 2002*). In amniotes, endochondral ossification involves both perichondral ossification and the invasion of the cartilage by blood vessels to deliver osteoblasts which deposit bone (*Hall, 2014*). In fish such as zebrafish, ossification involves the recruitment of osteoblasts to the surface of cartilage to secrete a perichondral collar of mineralized bone (*Hall, 2014*). Among vertebrates, the timing of the transition between endochondral growth and ossification is important in the determining size and shape of skeletal elements (*Nagata et al., 2011*; *Eames et al., 2011*; *Grünbaum, Cloutier & Vincent, 2012*; *Harrington, Harrison & Sheil, 2013*; *Arenas-Rodríguez, Rubiano Vargas & Hoyos, 2018*). This process is precisely regulated by expression of factors in chondrocytes and osteoblasts including *Ihh, Pthrp, Runx2 (Cbfa1),* and *Sp7* (*Osx*) (*Vortkamp et al., 1996*; *Komori et al., 1997*; *Nakashima et al., 2002*; *Maeda et al., 2007*). Loss of function of these factors can cause insufficient ossification, resulting in severe growth and patterning defects of the skeleton (*Quack et al., 1999*; *Wysolmerski et al., 2001*; *Gao et al., 2001*; *Valadares et al., 2014*).

Histone deacetylase 4 (HDAC4), a member of the class IIa group of HDACs (including HDAC4, 5, 7, and 9), has previously been demonstrated to be an important regulator of chondrocyte maturation and initiation of endochondral ossification in mice (*Vega et al., 2004*). HDAC4 lacks the ability to bind to DNA directly, but associates with other proteins to remove acetyl groups from the N-terminal of histone lysines, causing histones to condense, blocking access of transcription factors to DNA, resulting in transcriptional repression (*Haberland, Montgomery & Olson, 2009*). Class IIa HDACs are characterized by a carboxyl-terminal binding protein domain (CtBP), a Mef2 binding domain for binding the transcription factor MEF2C, sites for binding of the chaperone protein 14-3-3, and an HDAC domain (*Haberland, Montgomery & Olson, 2009*). MEF2C is a transcription factor which controls chondrocyte hypertrophy and bone formation by activating target genes such as *Runx2* (*Arnold et al., 2007*). When class IIa HDACs are unphosphorylated, they localize to the nucleus where they bind to MEF2C, and function to repress transcription of MEF2C-target genes (*Lu et al., 2000*; *Passier et al., 2000*; *McKinsey et al., 2000*; *Arnold et al., 2007*). When calcium/calmodulin protein kinase (CaMKII) and protein kinase D

(PKD) phosphorylate 14-3-3 and shuttle the Hdac4 into the cytoplasm, MEF2C becomes unbound to Hdac4, and can activate transcription of target genes (*Lu et al., 2000*; *Passier et al., 2000*; *McKinsey et al., 2000*). Through this process of interaction with MEF2C, HDAC4 delays the hypertrophy of chondrocytes within cartilage, controlling the timing and extent of ossification of endochondral bone by osteoblasts (*Vega et al., 2004*).

Zebrafish represent a useful model for studying mechanisms of chondrocyte maturation and the initiation of the cartilage ossification process as they develop cartilaginous elements as early as 60–72 hours post-fertilization (hpf) and commence perichondral ossification of these elements as early as 96 hpf (*Eames et al., 2013*). Compared with other vertebrates, zebrafish undergo the same cellular and genetic signaling pathways associated with skeletal ossification including chondrocyte hypertrophy, differentiation and matrix secretion by osteoblasts, including expression of factors associated with ossification such as *ihha, runx2a, runx2b, sp7, col1a2, col10a1,* and *osteonectin* (*Flores et al., 2004*; *Avaron et al., 2006*; *Li et al., 2009*). The similarity with other vertebrates such as mice make zebrafish a useful model to study genetic pathways associated with skeletal development and disease.

In this study, we describe two zebrafish lines with early frameshift mutations in *hdac4*. Mutant larvae from heterozygote intercrosses show an increase in ossification of ventral pharyngeal cartilage elements, and up-regulation of markers of ossification including *runx2a* and *runx2b*. A further enhancement of the excessive ossification defect is observed in maternal-zygotic mutants, indicating an early maternal contribution to skeletal patterning in zebrafish. Previously, we identified a potential role for *hdac4* in neural crest development and neurocranium formation in zebrafish (*DeLaurier et al., 2012*). Although this phenotype was not reproduced in any zygotic mutants, a profound loss of first pharyngeal arch facial structures was observed in a subset of maternal-zygotic mutants, indicating a function for maternal *hdac4* in neural crest development or formation of other anterior structures of the head or face. In conclusion, *hdac4* mutants reproduce aspects of the mouse *Hdac4* mutant for cartilage ossification and demonstrate a role for maternal *hdac4* in development of the anterior neural crest-derived facial skeleton in zebrafish. This mutant line may be a useful model, especially along with other reverse-genetic mutants for other class IIa Hdacs, to study the function of this class of epigenetic regulators in skeletal patterning and other developmental pathways.

## METHODS

### Zebrafish husbandry

AB strain wild-type (WT) zebrafish were originally obtained from the Zebrafish International Resource Center (ZIRC, Eugene, OR). Fish were reared and maintained at 28.5 °C on a 14 h on/10 h off light cycle. Fish were fed as previously described (*Wasden, Roberts & DeLaurier, 2017*). Maintenance and use of zebrafish followed guidelines from ZIRC, the Zebrafish book (*Westerfield, 2007*), and the Institutional Animal Care and Use Committee (IACUC) of the University of South Carolina Aiken (approval number 010317-BIO-01). Fish were staged using criteria for staging development up to 3 dpf (*Kimmel et al., 1995*). For 7 dpf samples, at least 95% of fish within a clutch had to show swim bladders in order to be included in analysis.

## Generation of CRISPR lines

The CRISPR/Cas9 procedure was based on previously described methodologies (*Hwang et al., 2013*). CHOPCHOP (http://chopchop.cbu.uib.no/) was used to design a guide RNA (gRNA) sequence targeting exon 5 of *hdac4* (ensembl ENSDART00000165238.3), 5′ of the Mef2c binding site (located in exon 6) and histone deacetylase domain, with minimal potential off-target binding (*Montague et al., 2014*; *Labun et al., 2016*). An *hdac4*-specific oligonucleotide was designed containing a 20 bp T7 promoter sequence, 20 bp of target sequence (GGAGCGTCATCGACAGGAGC), followed by a 20 bp scaffold overlap sequence as described (*Bassett et al., 2013*). This oligonucleotide was annealed to a scaffold oligonucleotide containing the tracrRNA stem loop sequence using Phusion PCR (New England Biolabs, Ipswich, MA, USA) to produce a 120 bp template. Template DNA was column purified (DNA Clean & Concentrator kit; Zymo Research, Irvine, CA) and was used to synthesize RNA using T7 polymerase (MAXIscript T7 Transcription kit; Thermo Fisher, Vitnus, Lithuania). Cas9 mRNA was synthesized from pCS2-nCas9n (Addgene, Cambridge, MA). The plasmid was linearized with *NotI*-HF (New England Biolabs, Ipswich, MA), column purified (Zyppy Plasmid Miniprep kit; Zymo Research, Irvine, CA, USA), and mRNA was synthesized (mMessenger mMachine SP6 kit; Thermo Fisher, Vitnus, Lithuania). Column-purified *hdac4* gRNA and nCas9n mRNA (RNA Clean & Concentrator kit; Zymo Research, Irvine, CA) were co-injected into one cell-stage embryos. Each embryo was injected with approximately 3nl of a 5ul mix containing *hdac4* gRNA (∼60 ng/microliter), nCas9n mRNA (∼160 ng/microliter), and phenol red as a marker. Unfertilized or dead embryos were removed from the dish at the end of the first day of injection and on subsequent days.

## Identification of founders and generation of mutant lines

At 36 hpf, approximately 20% of injected embryos (abnormal and normal-looking) were pooled into groups of 5 fish per tube, lysed using HotShot (*Truett et al., 2000*), and PCR was performed using genomic primers flanking the site of potential mutation. PCR products were gel-purified and digested using T7 endonuclease (New England Biolabs, Ipswich, MA, USA) to identify mismatched DNA indicating potential founder lines (*Hwang et al., 2013*). Siblings of fish (approximately 40 fish) with positive T7 results were reared to adulthood and used as founder ($F_0$) lines for subsequent experiments. Three $F_0$ fish demonstrated germ line transmission to offspring, and $F_1$ lines were generated from these founders by out-crossing founders to AB wild-types. Adult $F_1$ fish were identified as heterozygous carriers of potential mutations using PCR and T7 endonuclease digest on amputated tail DNA. Heterozygous $F_1$ fish were outcrossed to generate $F_2$ lines, and $F_2$ lines were intercrossed to produce homozygous mutants. PCR products from potential mutants and wild-type siblings were sequenced (Eurofins, Louisville, KY, USA) and genomic sequences were compared to wild-type siblings to identify mutations (Geneious, version 8, http://www.geneious.com) (*Kearse et al., 2012*). Mutations were confirmed by synthesis of cDNA from mRNA (RevertAid First Strand cDNA Synthesis kit; Thermo Fisher, Vitnus, Lithuania) and then amplified by PCR using an exon 3 forward primer 5′-gccactggaacttctcaagc-3′ and an exon 6 reverse primer 5′-gcagtggttgagactcctct-3′

(Tm = 58 °C × 40 cycles or Touchdown PCR, Tm = 72–65 °C × 15 cycles followed by Tm = 64.5 °C × 20 cycles). PCR products were column purified as described above and sequenced to confirm the mutation. Heterozygous $F_2$ and $F_3$ carriers of mutant alleles were intercrossed to produce wild-type, heterozygote, and mutant offspring. In order to test for the influence of maternal *hdac4* on development, maternal-zygotic mutants were generated using the $hdac4^{aik3}$ line by crossing homozygote mutant females with heterozygote males. Female mutants could not be generated using the $hdac4^{aik2}$ line, although mutant males survived.

## Genotyping adults and larvae

Amputated tails from adult fish, amputated tails from stained larvae, or whole larvae were genotyped using Hotshot lysis as described above. An 822 bp region of exon 5 spanning the site of mutation in *hdac4* was PCR-amplified using an intron 4–5 forward primer 5′-atgttctccctgtgttggtg-3′ and an intron 5–6 reverse primer 5′-gctgtatttccgctcatgtg-3′ (Tm = 58 °C, ×40 cycles). PCR products were run on a 2% agarose gel at 60V for 5–6 h to produce band separation sufficient to distinguish heterozygote (2 bands) fish from wild-type and mutant fish (both 1 lower and upper band, respectively).

## Alcian Blue and Alizarin Red histological stain

Alcian Blue and Alizarin Red staining to label cartilage and bone was performed as described (*Walker & Kimmel, 2007*). Briefly, 7 dpf zebrafish were fixed for 1 h in 2% paraformaldehyde/1X PBS, washed in 50% ethanol/$H_2O$, and stained overnight rocking at room temperature in 0.04% Alcian Blue (Anatech, Battle Creek, MI, USA)/0.01% Alizarin Red S (Sigma-Aldrich, St. Louis, MO, USA)/10 mm $MgCl_2$/80% ethanol. On day 2, fish were rinsed in 80% ethanol/10mM $MgCl_2$/$H_2O$ for several hours, then washed in 50% and 25% ethanol/$H_2O$, bleached using 3% $H_2O_2$/0.5% KOH for 10 min, then cleared in 25% glycerol/0.1% KOH, and incubated in 50% glycerol/0.1% KOH overnight, rocking at room temperature. Skeletal preparations were stored covered at 4 degrees Celcius.

## mRNA *in situ* hybridization

Single non-fluorescent and double fluorescent mRNA *in situ* hybridizations were performed as described (*Talbot, Johnson & Kimmel, 2010*; *Thisse & Thisse, 2014*), using probes for *hdac4, runx2a, runx2b, sp7,* and *sox9a* (*DeLaurier et al., 2010*; *Huycke, Eames & Kimmel, 2012*) using larvae from the $hdac4^{aik3}$ line. Fish were genotyped after staining. All mutant and wild-type larvae were used for confocal imaging and image analysis. No larvae were excluded from analysis with the exception of larvae that were damaged during processing or imaging. For mRNA *in situ* hybridizations of embryos from 2-cell to 90% epiboly stages, all embryos were collected from the same clutch of fertilized eggs on the same day, and fixed in 4% paraformaldehyde at specific stages. All embryos were pooled for *in situ* hybridization and developed for staining simultaneously. The reaction was stopped at the same point for all specimens. Specimens used for *in situ* hybridization were stored covered, in 1X PBS, at 4 degrees Celcius.

## Imaging, image analysis, and statistics: skeletal preparations

Alcian Blue and Alizarin Red stained specimens were dissected and flat mounted on microscope slides and imaged on a compound microscope. The right pharyngeal skeleton

was flat mounted for each specimen unless it was damaged or defective, in which case the left side was used. Among all genotyped fish, the extent of ossification of the ceratohyal was scored after removing sample identification and genotype information to ensure unbiased analysis. The region of ossification was scored for the evenness of the border of Alizarin red stain (regular border, irregular border), and for the approximate proportion of total area of the element that was stained (small area = less than approximately 20% or less of the total area, large area = greater than approximately 20% of the total area). In total, 94 fish were scored for the $hdac4^{aik2}$ line (39 wild-type, 23 heterozygotes, and 32 mutants), and 96 fish were scored for the $hdac4^{aik3}$ line (22 wild-type, 44 heterozygotes, and 30 mutants).

In order to quantitate the levels of ossification in wild-type, heterozygous, and mutant fish, the areas of flat mounted ceratohyal and hyosymplectic cartilages were measured using pixel area of cartilage and bone using ImageJ 1.51m9 (*Schneider, Rasband & Eliceiri, 2012*). In total, 93 fish were measured for the $hdac4^{aik2}$ line (38 wild-type, 23 heterozygote, 32 mutant) and 61 fish were measured for the $hdac4^{aik3}$ line (17 wild-type, 28 heterozygote, 16 mutant). The area of bone as a ratio of cartilage area was used a measure of ossification in statistical analysis (SAS 9.4; SAS Institute, Carey, NC, USA). Total area of the element was used as a method to control for differences in ossification due to subtle growth/stage differences between individual fish. For the $hdac4^{aik2}$ line, for the ceratohyal, and for the $hdac4^{aik3}$ line, for the hyosymplectic, measurements of "bone area" and "total area" met assumptions of normality and homogeneity of slope assumptions for analysis of covariance (ANCOVA). For the $hdac4^{aik2}$ line, for the hyosymplectic, and for the $hdac4^{aik3}$ line for the ceratohyal, there was no significant relationship between measurements of "bone area" and "total area" and so there was no reason to apply ANCOVA, and conventional ANOVA was applied. In the case of application of ANOVA, datasets met assumptions of normality and equal variance between comparison groups (Shapiro–Wilk normality test, Hartley's $F_{max}$-test, SAS 9.4; SAS Institute, Carey, NC, USA).

## Imaging, image analysis, and statistics: mRNA *in situ* hybridizations

Specimens used for fluorescent *in situ* hybridization were mounted ventral-side down on glass coverslips and imaged using an inverted Leica SPEII confocal microscope (Leica Microsystems, Buffalo Grove, IL, USA). In total, 21 4 dpf larvae stained using the *runx2a* probe were imaged (10 wild-type, 11 mutant), 17 4 dpf larvae stained using the *runx2b* probe were imaged (8 wild-type, 9 mutant), and 22 4 dpf larvae stained using the *sp7* probe were imaged (12 wild-type, 10 mutant). For *runx2a*, larvae from two stage-matched clutches were combined for analysis. For *runx2b* and *sp7*, only siblings from a single clutch were analyzed. Each sample was scanned to produce an approximately 80 z-slice stack. Confocal settings were kept exactly the same between imaging sessions. After scanning of all samples was complete, stacks were renamed (removing sample identification and genotype information) and randomized by a participant not involved in this study to ensure unbiased analysis of data. For all image stacks, gene expression was measured as the maximum length of expression (in microns) on the anterior and posterior margins of the right and left ceratohyal cartilages (Leica Application Suite X (LAS X) software 1.8.0.13370; Leica Microsystems). Measurements were recorded twice, on separate days, and data from

the two sessions were averaged to produce a final measurement for each region (i.e., left posterior, left anterior, right posterior, right anterior) of the ceratohyal for each sample. All four regions were averaged to produce a final average length of expression of each gene for each sample. Measurements for each genotype within each gene study were assessed for normality and equal variance (Shapiro–Wilk normality test, Hartley's $F_{max}$-test, SAS 9.4; SAS Institute, Carey, NC). All datasets met assumptions of normality and groups that were compared had equal variance, so ANOVA was performed.

### RNA-Seq data

We used the RNA-seq expression atlas data for zebrafish (https://www.ebi.ac.uk/gxa/experiments/E-ERAD-475/Results) to establish expression of *hdac4* transcripts per million transcripts (TPM) between zygote (1-cell) and 5dpf-stage larvae.

## RESULTS

### Hdac4 mutants have a frameshift lesion

Using CRISPR/Cas9 to target exon 5 of *hdac4* (Fig. 1A), we induced frameshift mutations in two individual fish that were used to generate mutant lines. The *hdac4*[aik2] allele has a 19 bp insertion three bases upstream of the protospacer adjacent motif (PAM) site associated with Cas9 binding and cleaving of DNA. The *hdac4*[aik3] allele has a 2 bp insertion, followed by retention of 7 bp of the wild-type sequence, followed by a 27 bp insertion one base pair upstream of the PAM site (Fig. 1B). In both cases, frameshifts were induced by insertion of nucleotides into exon 5, resulting in aberrant amino acids being added to the protein sequence (Fig. 1C). In both mutant lines the frameshift is predicted to cause the loss of the Mef2c binding domain and premature stop codons resulting in truncated proteins 174 aa (*hdac4*[aik2]) and 181 aa (*hdac4*[aik3]) in length (Fig. 1D). Frameshifts were detected in mutant cDNA compared to wild-type cDNA using primers spanning exons 3–6, and there was no evidence of splice variants or exon skipping detected on agarose gels for either mutant (Figs. 1E and 1F). In the case of *hdac4*[aik3], a larger band was detected along with the band of expected size (Fig. 1F, indicated by asterisk). This band was excised and sequenced and was found to have an identical sequence to the band of the expected size. We interpret that this larger band is the product of heterodimers of our PCR product or a slower running single-stranded DNA product and not a splice variant or other genomic feature within the mutant. In both the *hdac4*[aik2] and *hdac4*[aik3] lines, adult fish and embryos were genotyped using intronic primers spanning exon 5. The larger mutant band (841 bp *hdac4*[aik2], 858 bp *hdac4*[aik3]) can be distinguished from the wild-type band (822 bp), and heterozygotes show both bands (Figs. 1G and 1H). At 7 dpf and earlier, *hdac4*[aik3] mutants have no discernable external abnormalities compared to wild-type siblings (Figs. 1I and 1J). Mutant fish form swim bladders, feed, and grow normally into adult fish. Mutant fish from the *hdac4*[aik2] line also showed no overt external abnormalities compared to wild-type siblings (data not shown).

### Mutants have increased ossification of pharyngeal cartilage

Examination of Alcian Blue and Alizarin Red-stained specimens reveal that mutants from the *hdac4*[aik2] and *hdac4*[aik3] lines showed a greater extent of ossification of the ceratohyal

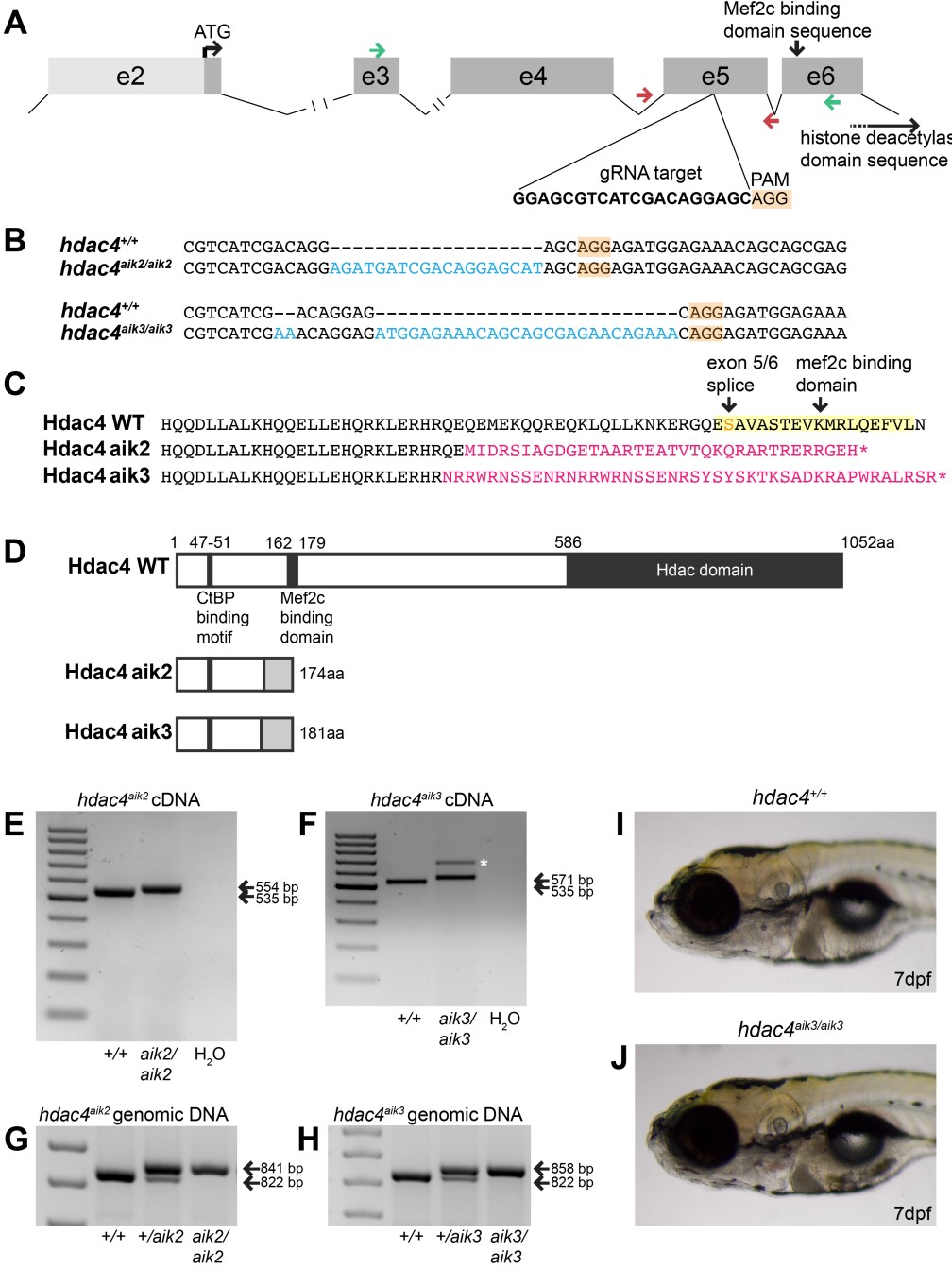

**Figure 1** **Overview of CRISPR strategy and generation of *hdac4* mutant lines.** (A) Genomic structure of *hdac4* showing gRNA target associated with the protospacer adjacent motif (5′-NGG, PAM) upstream of the Mef2c binding domain sequence. The histone deacetylase domain sequence is at the 3′ end of the gene. Green arrows indicate forward (exon 3F) and reverse (exon 6R) primers for RT-PCR and sequencing of cDNA. Red arrows indicate intronic genotyping primers (*hdac4* F6, *hdac4* R6) flanking exon 5. Intron 2/3 and 3/4 not to scale, indicated by hash marks. (B) Alignment of wild-type (*hdac4*+/+) with mutant cDNA sequence showing nucleotide insertions (blue) in *hdac4*aik2/aik2 and *hdac4*aik3/aik3 mutants in exon 5. (continued on next page...)

**Figure 1 (…continued)**
(C and D) Insertion of nucleotides results in reading frame shifts causing aberrant protein sequences (magenta in C, grey boxes in D), loss of the Mef2c binding domain (indicated in yellow in C), and premature termination of the protein sequence (asterisk in C indicates stop codon). (E and F) RT-PCR showing *hdac4* cDNA is spliced correctly in mutants and there is no evidence of splice variants. The wild-type cDNA product is expected to be 535 bp and mutant bands are 554 bp (E) and 571 bp (F). The larger band in *hdac4*[aik3] mutants (indicated by white asterisk in (F) was sequenced and determined to be identical to the lower band. (G and H) Genomic DNA samples were genotyped by PCR and show differences in band sizes indicating mutant (841 bp *hdac4*[aik2] mutant, 858 bp *hdac4*[aik3] mutant), wild-type (822 bp), and heterozygous fish (mutant and wild-type bands). (I and J) At 7 dpf, mutant *hdac4*[aik3] fish (J) have no apparent external abnormalities compared to wild-type siblings (I). $H_2O$ = negative control. 100 bp ladder.

cartilage compared to wild-type siblings (Figs. 2B–2G, *hdac4*[aik2] only shown in Figs. 2D–2G, *hdac4*[aik3] not shown). Wild-type fish had a smaller area of bone stain localized to the mid-shaft of the ceratohyal, usually appearing first at the dorsal margin of the cartilage and spreading ventrally, and this region of bone had even, regular, borders (Figs. 2D and 2E, see Figs. 2B and 2C for frequencies). Among mutants, a larger area of bone was observed at the mid-shaft of the ceratohyal, and this area of bone showed irregular borders (Figs. 2F and 2G, see Figs. 2B and 2C for frequencies). Among heterozygotes, the amount of bone area on the ceratohyal appeared as a mixture of the small and large areas observed in wild-types and mutants, and showed both regular and irregular borders (not shown, see Figs. 2B and 2C for frequencies). No other defects were detected in the pharyngeal skeleton or neurocrania in larvae analyzed at this stage.

For both the ceratohyal and hyosymplectic, the area of bone and the area of cartilage were measured, and the amount of ossification was calculated as the ratio of bone to cartilage present (Figs. 2H–2K). For the *hdac4*[aik2] line, ANCOVA revealed that the effect of genotype on the area of ossification of the ceratohyal was significant ($F = 4.01$; $df = 2,89$; $p = 0.0215$). Tukey's multiple comparisons showed that mutants had significantly more bone than wild-types ($p = 0.0057$), but heterozygotes were not significantly different from either mutants ($p = 0.1751$) or wild-types ($p = 0.2488$) (Fig. 2H). For the *hdac4*[aik3] line, ANOVA revealed that the effect of genotype on the area of ossification of the ceratohyal was also significant ($F = 7.77$; $df = 2,58$; $p = 0.001$). Tukey's multiple comparisons showed that mutants had significantly more bone than wild-types ($p = 0.0002$), that heterozygotes also had significantly more bone than wild-types ($p = 0.0134$), and that there was no significant difference between heterozygotes and mutants ($p = 0.0697$) (Fig. 2I). For the *hdac4*[aik2] line, ANOVA revealed no significant effect of genotype on area of ossification of the hyosymplectic ($F = 2.3$; $df = 2,90$; $p = 0.106$) (Fig. 2J). For the *hdac4*[aik3] line, ANCOVA also revealed no significant effect of genotype on area of ossification of the hyosymplectic ($F = 2.48$; $df = 1,56$; $p = 0.0928$) (Fig. 2K).

### *hdac4* is expressed in regions of the pharyngeal skeleton consistent with a role in cartilage maturation

mRNA *in situ* hybridization was used to detect *hdac4* transcripts in larvae at 72 hpf. Previously, we described the expression of *hdac4* in the ventral region of the developing pharyngeal skeleton at 72 hpf (*DeLaurier et al., 2012*), and here we show how specific regions

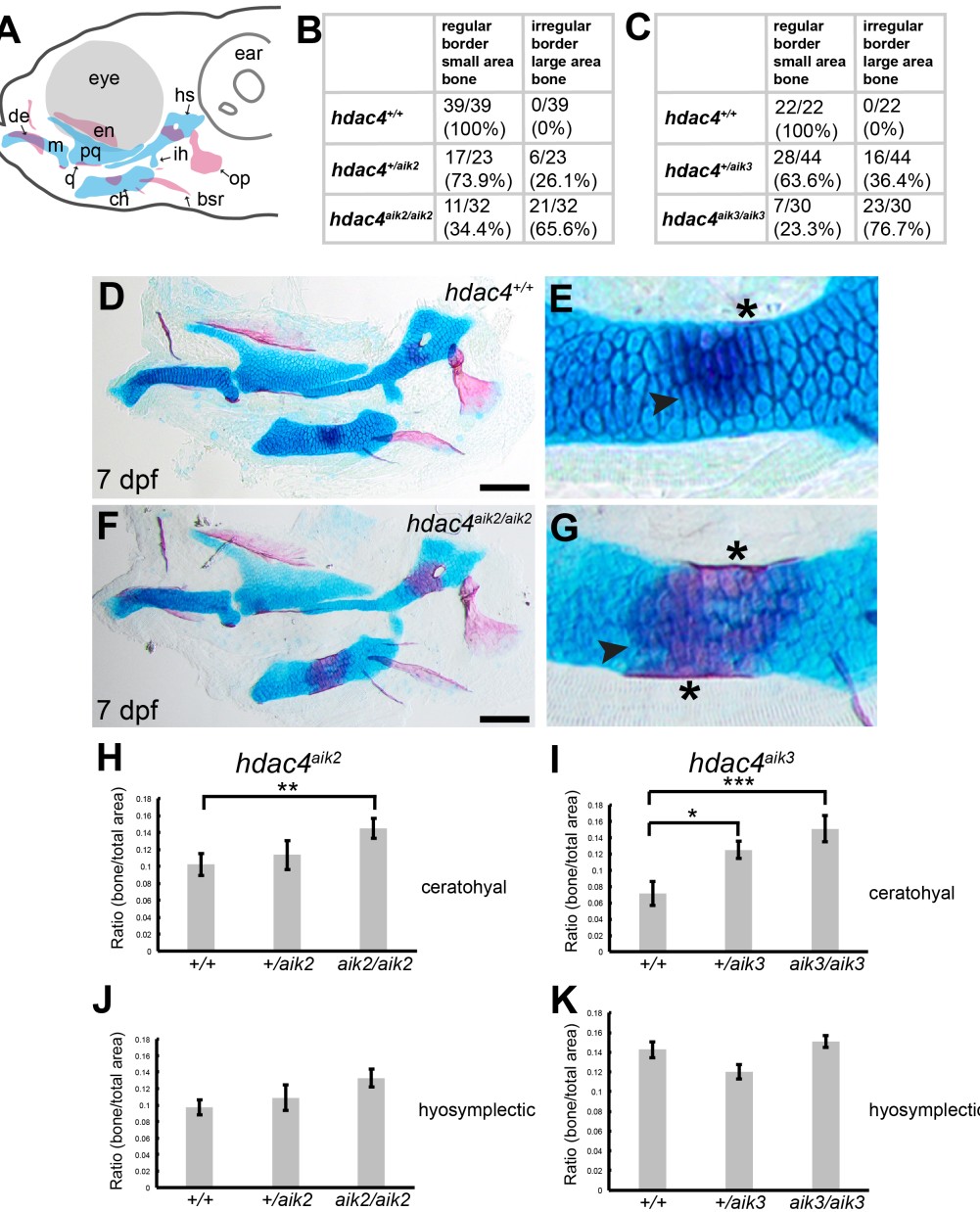

**Figure 2** **Analysis of ossification of the pharyngeal skeleton of *hdac4* zygotic mutants stained using Alcian Blue and Alizarin Red dyes.** (A) Schematic showing elements of the 7 dpf larvae pharyngeal skeleton, lateral view. Cartilage is indicated in blue, bone is indicated in red. (B and C) Total scores assigned to 7 dpf larvae scored for *hdac4*[aik2] and *hdac4*[aik3] lines, respectively. (D and E) Representative wild-type larval pharyngeal skeleton (D) and enlarged view (E) of ceratohyal showing small area (less than approximately 20% of total area of element) of bone formation, with even borders, indicated by arrow. Asterisk indicates perichondral bone collar. Lateral view. (F and G) Representative *hdac4*[aik2] mutant pharyngeal skeleton (F) and enlarged view (G) of ceratohyal showing large area (greater than approximately 20% of total area of element) of bone formation, with irregular borders, indicated by arrow. Asterisks indicate perichondral bone collar. Lateral view. (H and I) Bar graphs comparing ratios of bone to total area of the ceratohyal for *hdac4*[aik2] and *hdac4*[aik3] lines. (J and K) Bar graphs comparing ratios of bone to total area of the hyosymplectic for *hdac4*[aik2] and *hdac4*[aik3] lines. Bars in graph represent means and error bars are standard errors of means (SEM). (continued on next page...)

**Figure 2 (…continued)**
H, *hdac4*<sup>aik2</sup>: WT $n = 38$, mean = 0.102, SEM = 0.013; heterozygote $n = 23$, mean = 0.113, SEM = 0.017; and mutant $n = 32$, mean = 0.145, SEM = 0.0112. I, *hdac4*<sup>aik3</sup>: WT $n = 17$, mean = 0.071, SEM = 0.015; heterozygote $n = 28$, mean = 0.125, SEM = 0.011; and mutant $n = 16$ mean = 0.151, SEM = 0.016. J, *hdac4*<sup>aik2</sup>: WT $n = 38$, mean = 0.097, SEM = 0.009; heterozygote $n = 23$, mean = 0.109, SEM = 0.015; and mutant $n = 32$, mean = 0.133, SEM = 0.011. K, *hdac4*<sup>aik3</sup>: WT $n = 17$, mean = 0.142, SEM = 0.008; heterozygote $n = 28$, mean = 0.120, SEM = 0.007; and mutant $n = 16$, mean=0.151, SEM=0.006. Abbreviations: ch, ceratohyal; bsr, branchiostegal ray; de, dentary; en, entopterygoid; hs, hyosymplectic; ih, interhyal; m, Meckel's cartilage; op, opercle; pq, palatoquadrate; q, quadrate. $*p \leq 0.05$, $**p \leq 0.01$, $***p \leq 0.001$. Scale bar = 100 microns. Cartilage is stained blue (Alcian Blue), bone is stained red (Alizarin Red).

of expression are associated with sites of ossification of cartilage. By 72 hpf, expression is localized to regions of *sox9a*-expressing cartilage as well as in tissue surrounding cartilage and dermal bone elements. Co-expression of *hdac4* and *sox9a* was detected in the hyosymplectic, ceratohyal, and palatoquadrate cartilages (Figs. 3A–3H, indicated by arrows). In the case of the hyosymplectic and ceratohyal, co-expression of *hdac4* and *sox9a* was in regions that undergo ossification at later stages. *hdac4* was strongly expressed in the posterior pharyngeal arches, overlapping in the mid-region of each arch with a domain of *sox9a* expression in the ceratobranchial cartilage within each arch (Figs. 3I–3L).

## Mutants have increased expression of *runx2* factors

MEF2C is known to activate transcription of *Runx2*, a transcription factor that activates chondrocyte maturation (Arnold et al., 2007). *Runx2* in turn activates *Sp7*, a transcription factor associated with osteoblast differentiation (Nishio et al., 2006). We examined how loss of *hdac4*, which would lead to over-activity of Mef2c, affects levels of mRNA expression of these factors. Among teleost fish, a whole genome duplication event generated two copies of vertebrate *runx2, runx2a* and *runx2b* (Van der Meulen et al., 2005). As in previous studies, we observed differential patterns of expression of both genes, indicating that these genes have distinct functions in skeletal development in zebrafish (Figs. 4A–4H). At 4 dpf, *runx2a* and *runx2b* were expressed in the mid-shaft region of the ceratohyal cartilage and branchiostegal ray (Figs. 4B–4F), and also in the opercle (not shown). In general, *runx2a* expression was broader than *runx2b* expression at 4 dpf, encompassing more of the anterior and posterior lengths of the ceratohyal (Figs. 4B–4F). *runx2b* was expressed in the hyosymplectic at 4 dpf, but *runx2a* was not (not shown). Among mutants, the average length of *runx2a* expression in the ceratohyal was significantly increased compared to wild-types (Figs. 4A–4D, indicated by arrows, Fig. 4O; $F = 6.49$; $df = 1,19$; $p = 0.0196$). Expression of *runx2b* was also significantly increased in mutants compared to wild-types (Figs. 4E–4H, indicated by arrows, Fig. 4P; $F = 11.94$; $df = 1,15$; $p = 0.0035$). At 4 dpf, in both wild-types and mutants, *sp7* was expressed in a small patch of cells on the posterior margin, and less frequently also on the anterior margin, of the mid-shaft of the ceratohyal in both wild-type and mutant larvae (Figs. 4I–4L, indicated by arrows). Analysis of the average length of *sp7* expression revealed no significant differences between mutants and wild-types (Fig. 4Q, $F = 0.0001$; $df = 1,20$; $p = 0.99$). Schematic diagrams showing the differences in patterns of expression of *runx2a, runx2b,* and *sp7* in wild-type vs. zygotic mutant larvae at 4 dpf are shown in Figs. 4M–4N.

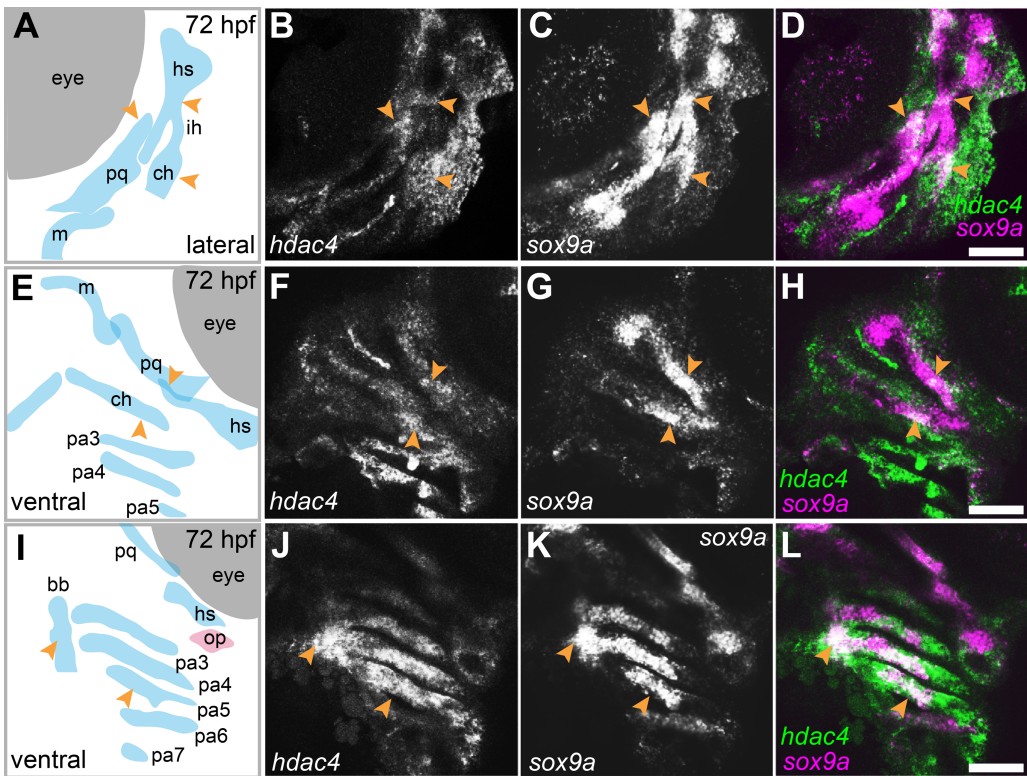

**Figure 3** **Expression of *hdac4* and *sox9a* mRNA in the pharyngeal skeleton of wild-type embryos at 72 hpf, detected using mRNA *in situ* hybridization.** (A) Schematic of skeletal elements, lateral view for B–D, blue indicates cartilage. (B–D) expression of *hdac4* and *sox9a*, arrows indicate expression in hyosymplectic, ceratohyal, and palatoquadrate cartilages. (E) Schematic of skeletal elements, ventro-lateral view for F–H. (F–H) expression of *hdac4* and *sox9a*, arrows indicate expression in ceratohyal and hyosymplectic cartilages. (I) Schematic of skeletal elements, blue indicates cartilage, red indicates bone, ventral view for J–L. (J–L) expression of *hdac4* and *sox9a*, arrows indicate expression in the posterior pharyngeal arches. Abbreviations: bb, basibranchial; ch, ceratohyal; hs, hyosymplectic; ih, interhyal; m, Meckel's cartilage; op, opercle; pa3-7, posterior pharyngeal arches 3-7; pq, palatoquadrate. Scale bar = 50 microns.

## Maternal-zygotic mutants have increased ossification of the pharyngeal skeleton and defects in the anterior facial region

In order to examine the role of a maternal contribution of *hdac4* to development, we performed analyses of skeletal phenotypes in mutants and heterozygotes generated from maternal mutants crossed with heterozygote males. At 7 dpf, a subset of maternal-zygotic mutants and heterozygotes showed a prominent increase in cartilage ossification compared to wild-types (non-sibling controls) (Figs. 5A, 5C–5E, 5G–5I). In total, 40.0% (16/40) of maternal-zygotic mutants and 40.0% (16/40) heterozygotes had evidence of premature or excessive ossification (Fig. 5F), and resembled ossification patterns present in 12 dpf wild-type fish (Fig. 5B). Increased ossification in mutants and heterozygotes included formation of an anguloarticular bone associated with the Meckel's cartilage (Figs. 5B–5D), an enlarged quadrate (Figs. 5C and 5E), a ventral hypohyal element associated with the ceratohyal (Figs. 5C and 5E), and ossification of the symplectic cartilage (Fig. 5C, indicated

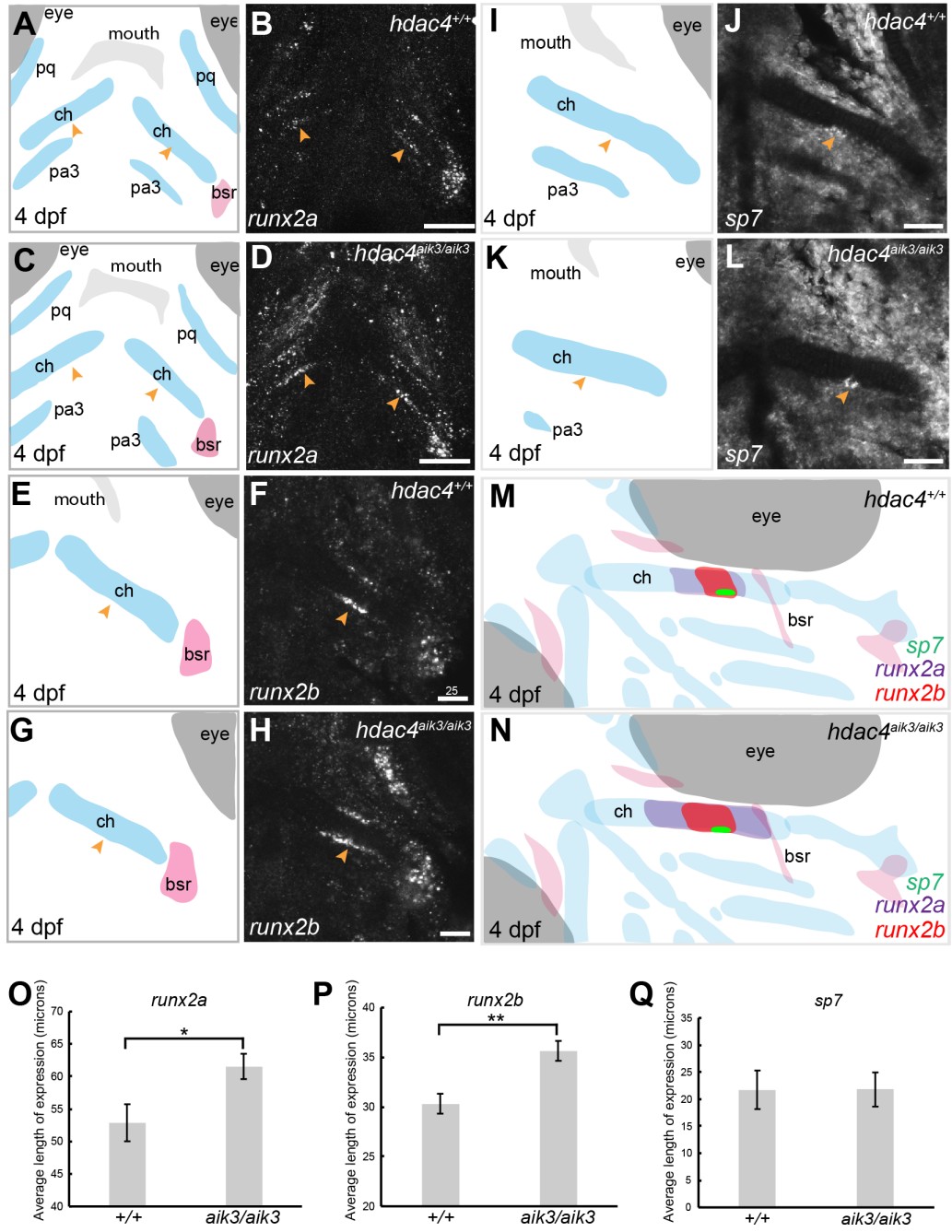

**Figure 4** **Expression of *runx2a*, *runx2b*, and *sp7* mRNA in the pharyngeal skeleton of wild-type and mutant larvae at 4 dpf, detected using mRNA *in situ* hybridization using the *hdac4*[aik3] line.** (A, C, E, G, I, K) Schematic views of skeletal elements, ventral view, blue indicates cartilage, red indicates bone. (B and D) expression of *runx2a* in wild-type (B) and mutant (D), arrows indicate expression at the posterior margin of the ceratohyal. (F and H) expression of *runx2b* in wild-type (F) and mutant (H), arrows indicate expression at the posterior margin of the ceratohyal. (J and L) expression of *sp7* in wild-type (J) and mutant (L), arrows indicate expression at the posterior margin of the ceratohyal. M and N, schematic showing overlapping domains of expression of *runx2a*, *runx2b*, and *sp7* in wild-type (M) and mutant (N) larvae. (continued on next page...)

**Figure 4 (…continued)**
(O–Q) Average length of expression, bars in graph represent means and error bars are standard errors of means (SEM); *runx2a* (wild-type $n = 10$, mean = 52.9, SEM = 2.86; mutant $n = 11$, mean = 61.55, SEM = 1.95), *runx2b* (wild-type $n = 8$, mean = 30.33, SEM = 1.34; mutant $n = 9$, mean = 35.65, SEM = 0.82), *sp7* (wild-type $n = 12$, mean = 21.73, SEM = 3.65; mutant $n = 10$, mean = 21.78, SEM = 3.15). Abbreviations: ch, ceratohyal; bsr, branchiostegal ray; pa3, pharyngeal arch 3; pq, palatoquadrate. *$p \leq 0.05$, **$p \leq 0.01$. B, D, scale bar = 50 microns, F, H, J, L, scale bar = 25 microns. Note that anterior and posterior lengths were at measured at different positions within the z-stack, and not all maximum lengths (anterior, posterior, right, and left ceratohyal) are represented in the sections shown in B, D, F, and H.

by asterisk). Among these maternal-zygotic mutants and heterozygotes, ossification of the mid-shaft of the first or first and second ceratobranchial cartilages was detected, which is also normally observed in wild-type larvae at 12 dpf (Figs. 5D, 5E, 5G–5I).

## Maternal-zygotic mutants have defects in the development of first pharyngeal arch skeletal elements and the anterior neurocranium

Among maternal-zygotic mutants, 25.9% (15/58) had defects of the first pharyngeal arch skeleton, including a shortened face (Figs. 6A, 6B and 6G) and loss of one or both Meckel's cartilages and the anterior portion of the palatoquadrate cartilage, with retention of a small remnant of the entopterygoid and quadrate (Fig. 6C, see Figs. 2A and 2D for reference). Among maternal-zygotic heterozygotes, 10.2% (6/59, Fig. 6G) also had loss of first pharyngeal arch structures. In the case of larvae with a loss of first arch cartilages, posterior second arch and more posterior arch structures including the ceratohyal, branchiostegal ray, and opercle were present (Fig. 6C). Some maternal-zygotic mutants and heterozygotes had defects in the neurocranium cartilage (the primary palate in fish), including clefts, holes, and shortening of the anterior portion of the element (Fig. 6D representative heterozygote with normal neurocranium, Figs. 6E and 6F heterozygote and mutant with neurocranium defects). In total, 30.8% (16/52) of maternal-zygotic mutants, and 15.2% (7/46) of maternal-zygotic heterozygotes had neurocranium defects (Fig. 6G). Most neurocranium defects occurred in fish that also had loss of first arch structures (11/16, 68.8% of maternal-zygotic mutants, and 5/7, 71.4% of maternal-zygotic heterozygotes).

## *hdac4* transcripts are detected in embryos younger than 50% epiboly

Based on the RNA-seq expression atlas data for zebrafish, the highest level of *hdac4* mRNA expression (relative to total transcripts) is between cleavage (2 cell) and blastula dome (over 1K cell stage), followed by relatively low expression of transcripts up to 3 dpf (larval protruding mouth), after which transcript levels increase (Fig. 7A). mRNA *in situ* hybridization experiments detected high levels of *hdac4* transcripts in wild-type embryos 512 cells or younger (Figs. 7B–7D, stages in between those presented also showed comparable expression). By 75–90% epiboly, *hdac4* expression was either not detectable, or expressed at very low levels compared to younger stages (Fig. 7E).

## DISCUSSION

In this study, we generated two novel zebrafish lines with germ line mutations near the start of *hdac4*. Both lines have insertions that cause frameshifts and premature stop codons,

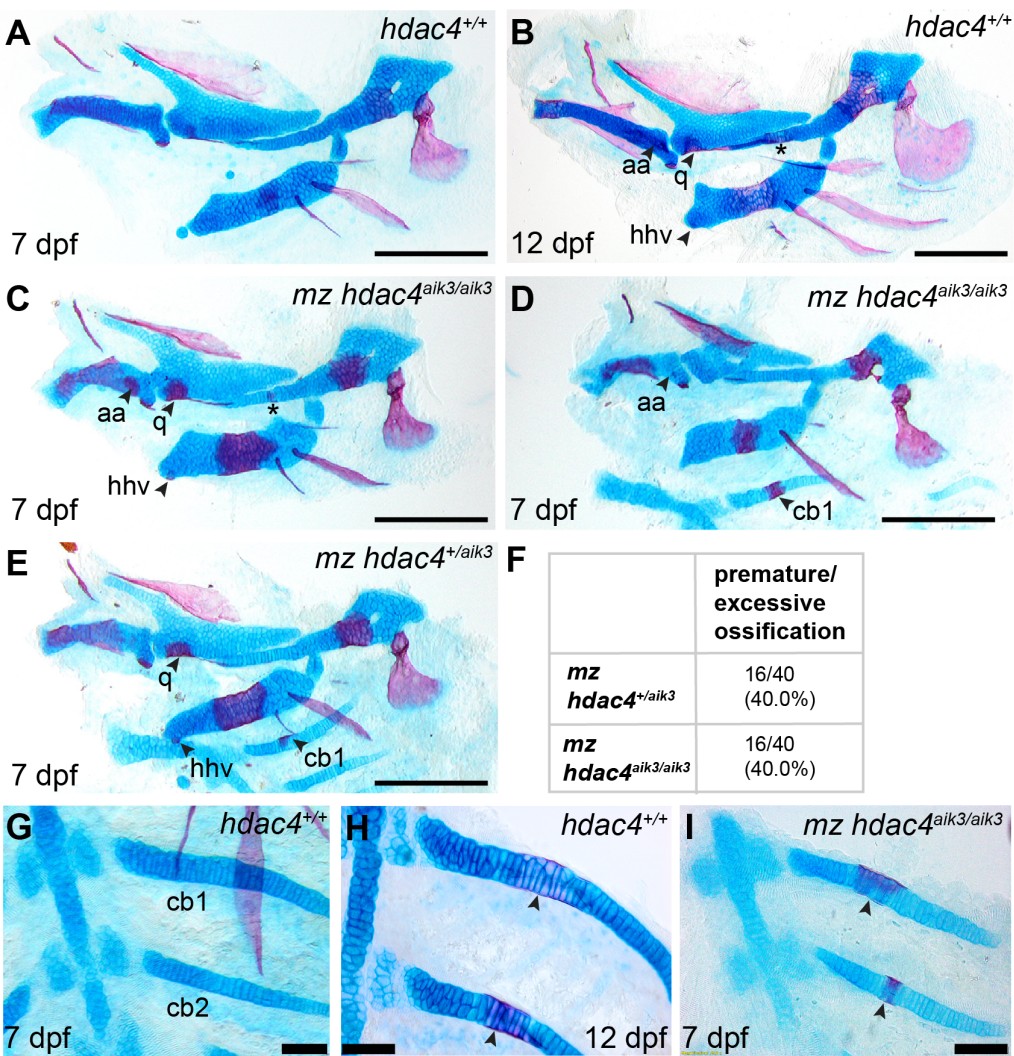

**Figure 5** **Analysis of maternal-zygote mutant and heterozygote cartilage ossification using specimens from the *hdac4*[aik3] line, stained using Alcian Blue and Alizarin Red dyes.** (A) Wild-type (non-sibling) larval pharyngeal skeleton at 7 dpf, lateral view. (B) Wild-type (non-sibling) larval pharyngeal skeleton at 12 dpf, lateral view. (C–E) maternal-zygotic mutant (C and D) and heterozygote (E) pharyngeal skeletons at 7 dpf showing premature ossification of the anguloarticular bone, ventral hypohyal, and ossification of the symplectic of the hyosymplectic (indicated by asterisk), lateral views. (F) Total scores of maternal-zygotic mutants and heterozygotes for premature/excessive ossification defects. (G and H) Wild-type posterior pharyngeal arches 3 and 4 at 7 dpf (G) and 12 dpf (H), including ossification of first and second ceratobranchial cartilages at 12 dpf, indicated by arrows, ventral views. (I) Maternal-zygotic mutant showing ossification of the first and second ceratobranchial cartilages cartilages at 7 dpf, indicated by arrows, ventral view. Abbreviations: aa, anguloarticular; cb1,2, ceratobranchial 1 and 2; q, quadrate; and hhv, ventral hypohyal. A–E, Scale bar = 200 microns, G–I, Scale bar = 50 microns. Cartilage is stained blue (Alcian Blue), bone is stained red (Alizarin Red). See Fig. 2A for schematic of the pharyngeal skeleton at 7 dpf.

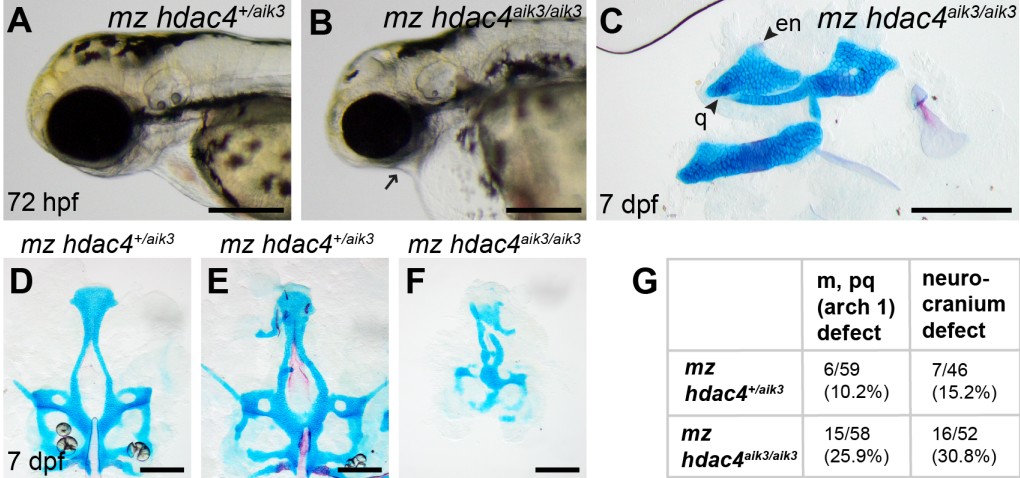

**Figure 6** **Analysis of maternal-zygote mutant and heterozygote first pharyngeal arch defects in fixed whole mount specimens and in specimens stained using Alcian Blue and Alizarin Red dyes.** (A) Maternal mutant heterozygote larvae, 72 hpf, lateral view. (B) Maternal-zygotic mutant, 72 hpf, lateral view. Arrow indicates facial reduction. (C) Maternal-zygotic mutant pharyngeal skeleton showing loss of first pharyngeal arch elements (see Figs. 2A and 2D for reference), 7 dpf, lateral view. (D and E) Neurocrania of maternal mutant-zygotic heterozygote larvae showing normal patterning (D) and defects (E), 7 dpf, ventral views. (F) Neurocranium of maternal-zygotic mutant, 7 dpf, ventral view. (G) Total scores of maternal-zygotic mutants and heterozygotes for first pharyngeal arch skeletal defects and neurocranium defects. Abbreviations: en, entopterygoid; m, Meckel's cartilage; pq, palatoquadrate; q, quadrate. Scale bar, 200 microns. Cartilage is stained blue (Alcian Blue), bone is stained red (Alizarin Red).

which we predict causes truncated proteins with no functional Mef2c binding domain or Hdac domain. Although a CtBP domain may exist in a truncated form of the mutant protein, we do not think this is sufficient to preserve function of the Hdac4 protein as a repressor of ossification in the absence of the Mef2c and Hdac domains. PCR and sequencing of cDNA does not reveal any alternative splice forms of the transcript, so we do not believe there is evidence for alternate versions of this protein caused by exon skipping or alternative splicing (*Sharpe & Cooper, 2017*).

Analysis of zygotic mutants revealed a statistically significant increase in ossification of the ceratohyal cartilage in both lines examined. One line showed a trend towards increased ossification of the hyosymplectic (*hdac4ᵃⁱᵏ²*), although the increase in bone was not significant in mutants compared to wild-type siblings. Normally, the hyosymplectic is the first element to commence ossification at 4–5 dpf, followed by the ceratohyal by 6 dpf (*Eames et al., 2013*). In the case of the ceratohyal, ossification normally begins on the anterior margin of the mid-shaft of the element and extends posteriorly and along the length of the cartilage element until around 12 dpf when the proximal end of the ceratohyal ossifies to form the epihyal (*Cubbage & Mabee, 1996*; *Eames et al., 2013*). Previous studies in the *Hdac4* mutant mouse show precocious ossification of the endochondral skeleton, with particular enhancement of ossification of chondral rib elements and limb cartilages in newborns (*Vega et al., 2004*). In both the mouse and the zebrafish, ossification of elements is not ectopic, but rather reveals a premature onset of the ossification process. We detected

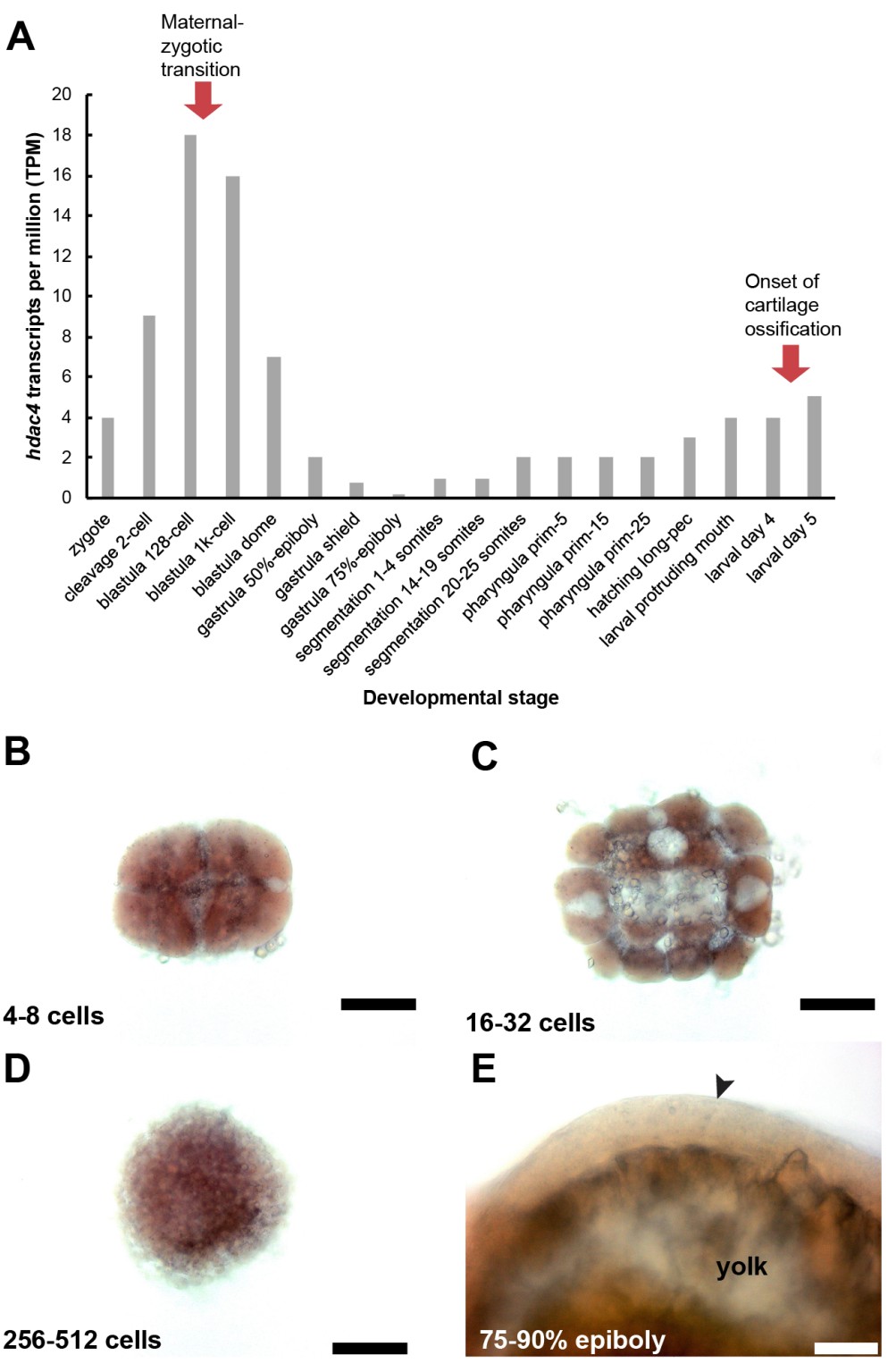

**Figure 7** Expression of *hdac4* mRNA transcripts at different stages of embryonic and larval development. A, RNA-seq mRNA levels of *hdac4* transcripts from zygotic to larval day 5 stage. Bars indicate numbers of *hdac4* transcripts per million transcripts at each stage. See http://www.ebi.ac.uk/gxa/experiments/ E-ERAD-475 for Expression Atlas data. B, Expression of *hdac4* mRNA at 4–8 cells, dorsal view. (continued on next page...)

**Figure 7 (…continued)**
C, Expression of *hdac4* mRNA at 16–32 cells, dorsal view. D, Expression of *hdac4* mRNA at 256–512 cells, dorsal view. E, Expression of *hdac4* mRNA at 75–90% epiboly, arrow indicates embryo dorsal to the yolk sac, lateral view. B-D, Scale bar = 200 microns, E, Scale bar = 50 microns.

that ossification was increased in heterozygote offspring (Fig. 2) compared to wild-types (*hdac4*$^{aik2}$ heterozygotes not significantly different to wild-types, *hdac4*$^{aik3}$ heterozygotes significantly different to wild-types). This phenotype in heterozygotes suggests that copies of both alleles of *hdac4* are required in zebrafish to repress ossification in the skeleton.

In mouse *Hdac4* mutants, *Runx2* expression is increased in cartilage, and is associated with the increase of endochondral ossification of elements (*Vega et al., 2004*). In zebrafish, orthologs of *runx2*, *runx2a* and *runx2b* are both expressed in embryonic and larval cartilage and bone, consistent with a function in ossification (*Flores et al., 2004*; *Li et al., 2009*; *Van der Velden et al., 2013*). At 48–96 hpf, *runx2a* is expressed in the maxilla, dentary, ceratohyal, opercle, and branchiostegal ray elements of the pharyngeal skeleton (*Flores et al., 2004*; *Li et al., 2009*; *Van der Velden et al., 2013*). At 48–96 hpf, *runx2b* is expressed in the ceratohyal, hyosymplectic, parasphenoid, entopterygoid, opercle, branchiostegal ray, and ceratobranchial cartilages of the third to seventh pharyngeal arches (*Flores et al., 2004*; *Li et al., 2009*; *Van der Velden et al., 2013*). We observed significantly increased expression of *runx2a* and *runx2b* in the ceratohyal of *hdac4*-mutant zebrafish compared to wild-types, which we speculate is due to a de-repression of Mef2c function leading to increased transcription of *runx2* genes. Phylogenetic analysis indicates that *runx2a* and *runx2b* are divergent paralogs, and thus may be regulated by different factors in zebrafish (*Van der Meulen et al., 2005*). In the case of our study, both *runx2a* and *runx2b* appear to be targets of repression by *hdac4*, and are not differentially affected by loss of *hdac4*. We did not detect any significant changes in *sp7*, a downstream target of *runx2*, in the ceratohyal of *hdac4* mutants compared to wild-types. Normally, *sp7* mRNA expression in the ceratohyal begins around 4 dpf (*Li et al., 2009*; *Hammond & Schulte-Merker, 2009*; *DeLaurier et al., 2010*). A possible reason why we did not detect significant differences between wild-types and mutants at this stage is because expression was still at early stages of initiation. Ideally, we would like to analyze *sp7* expression in older embryos (5 dpf or older). Unfortunately, in the present study we could not achieve adequate penetration of probe in whole mount samples at stages older than 4 dpf to assess *sp7* expression in the ceratohyal. Future studies will investigate *sp7* expression in older mutants and in maternal-zygotic mutants using a transgenic reporter for *sp7* (*DeLaurier et al., 2010*) or using mRNA *in situ* hybridization using cryosections.

Maternal-zygotic mutants and heterozygotes showed an enhancement of the ossification phenotype observed in zygotic mutants. At 7 dpf, these larvae had evidence of a ventral hypohyal at the distal portion of the ceratohyal, ossification of the symplectic cartilage of the hyosymplectic element, and ossification of the anterior first and second ceratobranchial cartilages. Ossification of the dorsal hypohyal occurs by 10 dpf, and ossification of the symplectic and first and second ceratobranchial cartilages occurs around 12–13 dpf (*Cubbage & Mabee, 1996*; *Eames et al., 2013*). The presence of ossification of these elements

as early as 7 dpf indicates that the ossification program is accelerated in maternal-zygotic mutants and heterozygotes. Among maternal-zygotic mutants and heterozygotes, the presence of an anguloarticular bone associated with the Meckel's cartilage was also observed. In our hands using Alcian Blue and Alizarin Red staining, and in other studies, this element is normally detected in wild-type fish by 12 dpf (*Cubbage & Mabee, 1996*). Other, more sensitive confocal imaging of Alizarin Red fluorescence has detected this element as early as 8 dpf (*Eames et al., 2013*). Unlike the other elements showing premature ossification in maternal-zygotic mutants, this element is a dermal bone that forms without a cartilaginous precursor (*Cubbage & Mabee, 1996*). Other dermal bones appear to be unaffected in maternal-zygotic *hdac4* mutants, and dermal bones are reported to be unaffected in *Hdac4* mutant mice (*Vega et al., 2004*). We cannot explain why this particular dermal bone appears prematurely in *hdac4* mutants; however, as it forms on the surface of the Meckel's cartilage, anguloarticular precursor cells may be responding to signals from the underlying Meckel's cartilage to commence formation of bone. During preparation of this manuscript it was noted that the cartilage of maternal-zygotic mutants and heterozygotes with excessive ossification generally had weaker Alcian Blue stain, and chondrocytes appeared rounder and less well organized compared to wild-type controls (Figs. 5C and 5D). These differences in chondrocyte morphology and cartilage matrix indicates that Hdac4 may repress chondrocyte hypertrophy in zebrafish.

The profound increase in ossification in a subset of maternal-zygotic mutants and heterozygotes compared to zygotic mutants suggests that there is a maternal influence on the chondral ossification program in zebrafish through expression of *hdac4*. In zebrafish, the maternal-zygotic transition (MZT) commences around 2 hpf (128 cells) where maternal transcripts are degraded and the first waves of zygotic transcripts are generated (*Tadros & Lipshitz, 2009*). Since the high levels of *hdac4* mRNA in embryos prior to 128 cells stage can only be maternal transcripts, we believe this indicates that there is a role for maternal *hdac4* in development. From our experiments, it is unclear how early maternal *hdac4* may be affecting ossification of cartilage several days post-fertilization. However, it is possible that maternal Hdac4 protein is still present in cells several days following the MZT and can influence ossification, or alternately, maternal *hdac4* may establish an epigenetic environment or signaling cascade in early embryos which has consequences on downstream skeletogenesis. A recent study has detected maternally-derived proteins in larvae up to at least 10 dpf (*Boer et al., 2015*), so it is possible that maternally-derived *hdac4* may regulate ossification processes in zebrafish up to 7 dpf. Future experiments will establish the levels of maternal *hdac4* transcripts or protein in zebrafish embryos, and we will examine the function of maternal transcripts or proteins on skeletal development in larvae.

Previously, we reported that morpholino knockdown of *hdac4* causes loss of neural crest and neurocranium defects in zebrafish (*DeLaurier et al., 2012*). Our zygotic mutants do not show any evidence of this phenotype. However, approximately one quarter of *hdac4* maternal-zygotic mutants show anterior neurocranium defects and a loss of first pharyngeal arch cartilages consistent with a role for *hdac4* in neural crest development. This phenotype is strikingly similar to a phenotype described for the zebrafish maternal-zygotic

mutants for fascin1a *(fscn1a)*, which have abnormalities in filopodia of a subset of cranial neural crest cells, causing migration defects resulting in a loss of the Meckel's cartilage and anterior palatoquadrate cartilage (*Boer et al., 2015*). Intriguingly, only approximately 20% of maternal-zygotic *fscn1a* mutants show this phenotype, similar to the 25.9% penetrance observed in *hdac4* maternal-zygotic mutants. In the case of *hdac4*, the proportion of approximately one quarter of maternal-zygotic mutants with defects suggests there may be effects of other unknown loci, modifiers, or interacting genes that are influencing the phenotypic outcome of loss or reductions of *hdac4* on neural crest development.

The observation in the previous study (*DeLaurier et al., 2012*) that knockdown of *hdac4* affects neural crest patterning is complicated to explain in the context of the neural crest defect detected in maternal-zygotic *hdac4* mutants. In the case of the *fscn1a* mutant study, a phenotype similar to *fscn1a* maternal-zygotic mutants was observed using a morpholino knockdown approach (*Boer, Jette & Stewart, 2016*). In this study, detailed analysis revealed that the *fscn1a*-morphant phenotype was not identical to the mutant, leading the authors to conclude that morpholinos can induce non-specific artifacts in neural crest cell migration and survival, which can be misinterpreted as a specific neural crest phenotype. Although *Boer, Jette & Stewart (2016)* used a translation-blocking morpholino (thus targeting maternal and zygotic transcripts) and (*DeLaurier et al., 2012*) used a splice-blocking morpholino (targeting only zygotic transcripts) we speculate that the defects observed in *hdac4*-morphants were due to sensitivity of cranial neural crest cells to morpholinos causing non-specific defects cell migration and survival, producing a phenotype of loss of the anterior neural crest-derived pharyngeal skeleton. Intriguingly, a recent study using a translation-blocking morpholino targeting *hdac4* reports partial clefting of the orofacial region of injected zebrafish (*Rothschild et al., 2018*). We believe the phenotype reported in *DeLaurier et al. (2012)* is the result of morpholino-induced artifacts producing non-specific defects to cell migration and survival. We believe this is the only explanation for the presence of a neural crest phenotype in *hdac4*-morphants (reported in *DeLaurier et al., 2012*) and a lack of a neural crest phenotype in zygotic mutants. Furthermore, because the morpholino could not target maternal transcripts, we believe the maternal contribution was unaffected in morpholino knockdowns, further reinforcing the idea that the *hdac4* morphant phenotype reported by *DeLaurier et al. (2012)* may be the product of a non-specific artifact. Despite the complicating evidence of morpholino data in the light of the current study, we believe that the mutant line presented here reveals a novel function for maternal *hdac4* in neural crest development.

Based on our findings in maternal-zygotic *hdac4* mutants and heterozygotes, our future experiments aim to characterize the cellular defect in maternal-zygotic mutants. We will establish if the loss of anterior facial structures is due to a failure of cranial neural crest cell migration or another defect such as specification of neural crest cells, or defects in the patterning of neural crest-derived skeletal elements. We will also establish the levels of specifically maternal *hdac4* in zebrafish embryos and we will examine the function of maternal transcripts or proteins on neural crest migration and patterning of skeletal elements in embryos and larvae.

## CONCLUSIONS

In conclusion, this study shows that mutation of *hdac4* in zebrafish causes premature ossification of the pharyngeal skeleton, consistent with previous findings in the mouse (*Vega et al., 2004*), indicating a conserved function for *Hdac4* among vertebrates. Mutants have increased expression of the transcription factors *runx2a* and *runx2b*, activators of the skeletal ossification program, which we speculate are upregulated in response to increased activity of Mef2c through loss of *hdac4*. Maternal-zygotic crosses, along with RNA-seq analysis and detection of high levels of *hdac4* mRNA in 512-cell and younger embryos indicate that maternal *hdac4* is an important contributor to embryonic and larval development. A subset of maternal-zygotic mutants and heterozygotes show an enhancement of the premature ossification defect observed in zygotic mutants. A subset of maternal-zygotic mutants and heterozygotes also show profound loss of the anterior facial skeleton, including loss of the anterior neurocranium and first pharyngeal arch-derived elements. Although this study did not determine the underlying mechanism for loss of neural crest-derived skeletal elements in maternal-zygotic mutants and heterozygotes, the striking similarity of the *hdac4* mutant with the *fscn1a* mutant (*Boer et al., 2015*) suggests loss of *hdac4* may be associated with a migration defect in a subset of cranial neural crest cells.

## ACKNOWLEDGEMENTS

We would like to thank Jared Talbot, Nathan Hancock, and Alec Jones for providing feedback on drafts of this manuscript, Derek Zelmer and Virginia Shervette for assistance with the statistical analysis, and the Busch-Nentwich lab for providing RNA-seq data.

### Funding

This work was supported by the NIH/NIGMS grant to SC INBRE P20GM103499, funding from University of South Carolina RISE, ASPIRE-I, and ASPIRE-III awards to April DeLaurier, and start-up funds from University of South Carolina Aiken. The funders had no role in study design, data collection and analysis, decision to publish, or preparation of the manuscript.

### Grant Disclosures

The following grant information was disclosed by the authors:
NIH/NIGMS: INBRE P20GM103499.
University of South Carolina RISE, ASPIRE-I, and ASPIRE-III awards.
University of South Carolina Aiken.

### Competing Interests

The authors declare there are no competing interests.

## Author Contributions

- April DeLaurier conceived and designed the experiments, performed the experiments, analyzed the data, contributed reagents/materials/analysis tools, prepared figures and/or tables, authored or reviewed drafts of the paper, approved the final draft.
- Cynthia Lizzet Alvarez performed the experiments, analyzed the data, approved the final draft.
- Kali J Wiggins performed the experiments, approved the final draft.

## Animal Ethics

The following information was supplied relating to ethical approvals (i.e., approving body and any reference numbers):

South Carolina Aiken IACUC approved this research (010317-BIO-01).

## DNA Deposition

The following information was supplied regarding the deposition of DNA sequences:

Mutant sequence data are provided in the Supplemental Files.

## Data Availability

The raw data are provided in the Supplemental Files.

## Supplemental Information

Supplemental information for this article can be found online at http://dx.doi.org/10.7717/peerj.6167#supplemental-information.

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
