# Peer review of "hdac4 mediates perichondral ossification and pharyngeal skeleton development in the zebrafish"

_PeerJ, doi:10.7717/peerj.6167_

## Round 0.1 · original submission · Major Revisions

Thank you for submitting your work to PeerJ. While all three reviewers were positive about the contributions of this study, each have a number of recommendations that I invite you to respond to with a resubmission of your manuscript. The reviewers’ recommendations would enhance the presentation of your study as well as strengthen the support for your conclusions. Due to the extent of requested changes in presentation and suggestions for additional analyses I am recommending that major revisions be made. Please feel free to contact me if you have any questions as you make revisions and prepare a response to the reviewers.

Reviewer 1 ·

Basic reporting

In this manuscript, DeLaurier demonstrated that disruption of hdac4 caused increase of ossification in the Ch cartilages, with enhanced expression of runx2a and sp7. Also, maternal-zygotic mutants showed enhanced ossification than the zygotic only mutants. This study revealed a role of hdac4 in cartilage ossification. The manuscript is well written, but still needed further improvement.

Major issues:

1. In Fig3, The expression of hdac4 at 72hpf is very different from the previous reported paper (DeLaurier et al., 2012). The in situ seems to be in bad quality, it is not convincing about the colocalization of sox9a.

hdac4 and sox9a are highly co-expressed in the bb and cb3-5. Are there any defects in these skeletal elements?

2. Is hdac4 highly expressed in the first arch? The MZ mutants show severe defects of 1st arch cartilages.

3. Are the zygotic hdac4 mutants showing any defects in the neurocranium?

4. In Fig 4, authors claimed that runx2a and sp7 were increased in the mutants. Authors need to quantify the changes and reveal the ratio of embryos showing the defects.

5. In Fig 5, MZ hdac4 mutants show more severe defects than the zygotic mutants. Authors should detect the bone marker runx2a, runx2b and sp7 expression in the MZ mutants.

Experimental design

no comment

Validity of the findings

no comment

Reviewer 2 ·

Basic reporting

The manuscript is well written, follows an appropriate structure for reporting of scientific information and the figures are of extremely high quality to convey the information being presented. The information provided in the background and discussion is thorough and provides adequate context to interpret the results presented.

However, in general, the manuscript could benefit from some relatively minor modifications to improve the readability for a more general audience (outside the zebrafish/developmental/craniofacial/skeletal fields). Most notably, this includes clarifying methodologies used in each of the figures and double checking gene/protein nomenclature, specifically for cited mouse work. These suggestions are indicated with an asterisk (*) below.

Specific suggestions (in order of appearance) are as follows:
BR1. Line 145: Add “The” at start of the sentence so it reads, “The plasmid...”.

BR2. Line 204: would prefer use of hyosymplectic over hyomandibula (if they can be used interchangeably at this developmental stage) for consistency with main body of the text and figures. There are no other appearances of hyomandibula in the main body of the text.

BR3. Line 236 to 238: Reword the fish have “no discernible abnormalities” for clarity. The fish appear normal to the eye, but a phenotype can certainly be discerned at this stage as demonstrated in Figure 2. The allele for which this statement is being made is also not specified (aik3, based on the Figure, but this should be mentioned in the main text).

*BR4. Line 244: briefly reference the method used to detect cartilage and ossifying elements. One sentence is sufficient. (For example: add “At 7dpf, flat mounted sections of fixed/fresh (?) tissue stained with... were analyzed for bone defects” etc.) At a minimum, include some description of staining and mounting or mention of the methodology used in the figure legend.

*BR5. Line 279: Again, a mention of the method used (in situ) to assay gene expression would be preferred in the main body, but at a minimum in the legend for Figure 3. This is especially important because zebrafish are still transparent at this stage (72hpf) and transgenic fish are often used to detect gene expression/localization. One phrase/sentence would be sufficient.

*BR6. Line 291 to 292: Double check nomenclature (specifically for mouse) for all genes/proteins to help clarify when talking about genes/transcripts or proteins for different species. Except in rare cases, mouse proteins should be in all caps. (Please check entire manuscript, however, other instances to check include lines 374, 459 and 461.

BR7. Line 299 to 312: Panels in Figure 4 are referred to out of order in the text. Suggestions:
- a general reference to Fig. 4 in line 299, or “(Fig. 4 A-H)”
- line 303, “(Fig. 4 A-D, indicated by arrows)”
- line 306, remove reference to figure
- line 307, “(Fig. 4 E-H, indicated by arrows)
- line 310, add “(Fig. 4 I-L, indicated by arrows)
- line 311, remove reference to figure
- add sentence at the end of the paragraph (~line 312) describing Fig. 4 M and N; for example: “A summary of the changes seen in runx2a, runx2b and sp7 are depicted in Fig. 4 M and N.” This also provides an opportunity to restate the findings of Fig. 4, if desired.

BR8. Line 321: would prefer something denoting the “enlarged quadrate” in Fig. 5B (arrow, asterisk, etc.) instead of, or in addition to, a reference to Fig. 2.

BR9. Line 474: check instance of mef2c-/-;mef2c+/-, missing a/b designations

BR10. Figure 2 legend: E’ there is only one arrow in the figure even though two are mentioned in the legend. Either omit reference to second “lower arrow” (not mentioned in the main text) or add arrow to the relevant important feature in figure. If adding a second arrow, would prefer use of “arrowhead” in place of “lower arrow” (or the other arrow, depending on relative location) to distinguish between the two shapes when/if arrow is added to figure so the reader can easily distinguish which element is being referenced as an “irregular border” vs. “spread of ossification to ventral aspect...”.

BR11. Figure 3 legend: In “A”, remove text, “red indicates bone”, as there is no red/bone in this schematic. Move this text to “I”. In “E” figure refers to ventral view, but legend says lateral. Also, see note BR5 about mentioning methodology used.

Experimental design

In general, the goal of the study is quite evident and the experiments used to assess the role of Hdac4 in zebrafish early larval perichondral ossification are appropriate and well-designed. There are a few major considerations, however, that I feel must be addressed by the authors prior to publication. These are as follows (in order of importance):

ED1. While the authors do a sufficient job convincing the reader that early ossification occurs in the Hdac4 mutant lines, it is difficult to assess the degree to which this is true. For example, in line 365 to 373, the comparison of mutants to more developed zebrafish is an important one. This point is well-made and the impact would be even greater if the authors could provide an image of a wild type, 12dpf (or similarly staged), fish prepared using their staining and tissue preparation method.
Again, in the discussion (see lines 406 to 408 and lines 410 to 412), it is difficult to assess the degree of precocious ossification alluded to in these statements without reference images using a similar method on older larvae. The craniofacial atlas referenced in Eames et al. 2013 shows some ossification of the ceratobranchial (cb) as early as 4dpf. If this statement is specific to a certain ceratobranchial bone (e.g. cb1/2 vs cb5) that distinction should be acknowledged. Similarly, the atlas denotes an anguloarticular bone (aa) is detectable as early as 6-8dpf (“Other bones visible here between 6 dpf and 8 dpf include the anguloarticular (aa) and quadrate (q) bones...”). Otherwise, the differences may be due to methodological or staging in these fish, which would be easily resolved by presenting comparison images of older larvae.

ED2. Somewhat related to ED1, the results presented in Fig. 2 and Fig. 5 (particularly Fig. 2B/C and Fig. 5G) would be easier to interpret and more impactful if data relating to developmental staging of these fish were available (e.g. swim bladder or eye size, see Parichy et al. Dev Dyn [2009] "Normal table of postembryonic zebrafish development: staging by externally visible anatomy of the living fish"). This would demonstrate that potential growth differences in these fish were considered (at a minimum) and/or controlled for (ideally), which seems relevant and especially important given that the phenotype in zygotic mutants is later described (lines 365 and 372 in the discussion) as appearing in a normal pattern as opposed to being ectopic.

ED3. The presentation of two CRISPR-generated alleles is a strength of the paper. However, it is unclear why some figures include data for both alleles, some for only aik2, and some for only aik3. A rationale of why certain allele(s) were chosen for each experiment (including the generation of maternal-zygotic mutants using the aik3 allele) would be appreciated.

ED4. In Figure 2, a more thorough explanation of criteria for classifying “mild” and “moderate” ossification patterns is needed. It is hard to distinguish the difference between the normal and “mild” phenotype shown in Fig. 1 D’ and Fig. 1 E’ (there also appears to be a small bone collar formation at the upper right area of ossification in D’ “the normal phenotype”, similarly sized to that designated by an asterisk in E’). Was the observer blinded during scoring? Were objective criteria used to classify patterns or subjective measures? Did scoring consider ossification of only the ceratohyal, the ceratohyal and the hyosymplectic, or both, or the entire tissue preparation?

Specific, minor considerations:
ED5. Line 190 to 192: A brief expansion of the histological methodology used would be welcome, as full-text to the reference cited is not widely accessible.

ED6. Line 258: Justification for normalizing to cartilage area should be included. If this is due to staging concerns, would another measure (independent of skeletal formation) be more reliable?

ED7. Line 316: A sentence describing the rationale for moving to maternal-zygotic mutants should be included.

Validity of the findings

Overall, the conclusions the authors make are supported by the data in the manuscript. A few changes relating to the rigor of statistical testing and the validity of statements made are below (in order of importance):

V1. Line 258 to 271: Justification for using a different statistical test on similar analyses should be given. If there is no justification, data should be re-analyzed using only ANOVA -or- ANCOVA.

V2. Line 355 to 359: Authors may need to adjust interpretation depending on revised statistical analysis.

V3. Line 344, and line 350 to 354: The prediction that the gene product for aik2/3 is a truncated protein is unfounded. The presence of amplified cDNA is not sufficient to suppose that a protein product is generated, or that the RNA is not subjected to NMD. Quantitative real-time PCR would be a much more reasonable experiment to assess relative cDNA expression in WT, hets and mutants for each allele. Of course, Western blots and/or immunostaining against an N-terminal portion of Hdac4 would be required to conclusively make this statement/prediction although antibodies for Hdac4 may not be readily available for zebrafish. It would be preferable to support this prediction with additional experiments or to remove these statements from the manuscript.

V4. Line 269: typo in the p-value, cannot be greater than 1.0

·

Basic reporting

The micrographs are of high quality and overall the imaging and figures are very nice.

The manuscript is clearly written for the most part. However, there are numerous examples of incorrect gene nomenclature usage. For example, on line 368 the authors refer to the mouse Hdac4 gene and it is italicized, on line 374 the authors discuss Hdac4 and Runx2 mouse genes and neither are italicized. There are numerous such incorrect and inconsistent nomenclature usages like on line 461 the zebrafish gene mef2ca is capitalized and it should be all lower case.

There are numerous statements that require references. For example the sentences ending on lines 56, 61, 64 and others need references.

The Mef2c gene is “myocyte enhancer factor 2c” NOT “myocyte enhancing factor 2c”

In my opinion, “intercross” should be used rather than “in-cross”.

Stylistically, the authors could improve the readability by avoiding the passive voice throughout.

Experimental design

The manuscript is within the aims and scope of the journal. Examining the functional role of hdac4 in zebrafish development with a genetic mutant is meaningful especially since the only loss of function phenotype reported thus far is a morpholino-based study which often do not phenocopy germline mutations. The methods are sufficiently described.

Validity of the findings

To strengthen the manuscript to "pass" the following points need to be addressed.

In order of importance:

1. The ossification phenotype in the zygotic mutants (Fig 2D-F) is very subtle. Moreover, the range of ossification reported to be due to hdac4 mutation can be observed in wild type animals at this stage (7dpf). In light of these facts, extra careful quantification is required. Specifically, the authors need to quantify bone blind to hdac4 genotype. Perhaps using the intrinsic fluorescence of Alizarin red in whole mount larvae from a heterozygous intercross prior to genotyping will speed analyses. Quantifying ossification BEFORE genotyping then determining if the ossification phenotype segregates with hdac4 genotype will lead to a much more convincing demonstration that hdac4 mediates perichondral ossification as the authors propose.

2. An additional concern with the data in Figure 2 is that the “moderate” phenotype in Fig 2F looks like an overall developmentally advanced individual based on a more developed branchiostegal ray set, a more well-developed interopercle and larger cartilage elements (eg palatoquadrate) than those shown in the control or “mild” in D and E respectively. These observations imply that the ‘moderate’ phenotype is an overall advanced individual rather than a phenotype specific to a certain bony element. This sort of stage variation is common even in wild-type larvae. Indeed, the authors report some wild type individuals with the ‘mild’ phenotype in their raw data, weakening the claim that this phenotype is due to hdac4 mutant genotype. Even in the MZ mutants the ossification phenotype appears to be within the normal range of ossification variation that is observed in wild-type animals.

3. Similarly, the subtle differences in runx2a and sp7 expression in figures 4B,D and 4J,L are within the variation seen among wild types. If the authors wish to make the point that these subtle expression differences are meaningful they should be backed up with quantitative analyses like qPCR. At a minimum, the fluorescence in situ signal from many individuals should be quantified BEFORE genotyping to determine if the in situ signal phenotype segregates with hdac4 genotype.

4. In Fig 5, the loss of neurocrania and first arch structures while the second arch appears completely wild type is both fascinating and convincing. This phenotype should be the centerpiece of this study. The low penetrance of such a severe phenotype is also very interesting and worth study. This manuscript would be dramatically strengthened by focusing on understanding the loss of arch structure phenotype, rather than the subtle ossification phenotype. Do the NCCs migrate normally? Is Arch morphology and size normal? Are there arch morphology differences between arch 1 and 2? Are patterning genes only disrupted in arch1 while arch 2 is wild type?

5. Does the MZ phenotype (Fig. 5) match the previously reported MO phenotype? It is difficult to imagine that the previous MO study was blocking maternally deposited transcript because splice-blocking MO were used (DeLaurier 2012). The similarities and differences between the MO and MZ mutant should be clearly fleshed out. An interesting experiment to better understand this phenotype an how it contrasts with the previously reported MO phenotype would be to inject the MO into MZ mutants.

6. It is certainly important to consider how hdac4 may be functioning during craniofacial development, i.e. through what target genes and interactions. However, this manuscript contains no evidence whatsoever that hdac4 functions via mef2c. Without any data, the mef2c/hdac4 interaction in this system is purely speculative and therefore any reference to an hdac/mef2 interaction should be relegated to the discussion and clearly stated as speculative. ie, removed from the 'results section' and the abstract. For example, the authors state their hypothesis in the abstract that “mutant larvae have an increase of expression of runx2a and sp7 in the ceratohyal cartilage, which we hypothesize is due to the de-repression of Mef2c through loss of Hdac4.” But then they do not test this hypothesis at all.

7. It is curious that the authors report an ossification phenotype in the hdac4 heterozygotes. Is there any reason to suspect that the alleles are dominant? Dominance needs to be addressed. If the authors suspect haploinsufficiency then this should be stated.

8. The authors argue that maternally deposited hdac4 functions in skeletal development. They could formally demonstrate that hdac4 mRNA is maternally deposited if they were to examine the amplified cDNA sequences in zygotic mutants obtained from heterozygous intercrosses. If any wild type sequences were found in cDNA generated from homozygous mutants they would unambiguously demonstrate that wild type hdac4 mRNA is maternally contributed in zygotic hdac4 mutatns. The authors already have the required PCR products to test this hypothesis (1E,F)

Additional comments

Clean mutagenesis, including multiple alleles, leading to an interesting craniofacial phenotype (5 J-M) make this project an important contribution to our understanding of craniofacial development. I strongly suggest shifting the focus of this work to the most robust (and unusual) phenotype rather than the subtle increase in ossification. At a minimum the ossification phenotype needs to be much more carefully quantified as described in the ‘validity of the findings’ section above.

Please find a marked up PDF with some candid comments that may help the authors edit this manuscript. Many of the comments are redundant with those in the review but some additional frank (non proofread) comments are provided which I hope will be helpful.

---

## Round 0.2 · Minor Revisions

Thank you for submitting your revised manuscript to PeerJ. All three reviewers were very positive about the changes made. Two of them had some additional minor recommendations, which I invite you to address in a second resubmission and rebuttal letter.

Reviewer 2 expresses some concerns with how your RT-PCR data are used to address the possibility of nonsense-mediated decay (NMD). I agree with the reviewer that the presence of bands resulting from RT-PCR might not indicate the amount of NMD taking place. I don’t believe that the size of the band would be diagnostic either. I concur that conclusions about NMD should be removed from the discussion, but am open to a defense of these claims.

The methods (page 10 line 188) mention RT-PCR of cDNA. Should this be written as “RT-PCR of mRNA”?

Reviewer 3 identifies some remaining issues with gene nomenclature. For the most part these have been corrected, but there do seem to be some remaining names that need to be edited using the conventions for mouse and zebrafish that you follow through most of the manuscript. I also agree that when mentioning the alleles you should note them once “hdac4^aik3” unless mentioning a homozygous mutant. There are a few other minor suggestions that should be responded to in your resubmission and rebuttal letter. Reviewer 3 does ask for one additional set of data on the phenotype of MZ or zygotic mutants at 12 dpf. I will leave it to your discretion whether these data should be added.

Reviewer 1 ·

Basic reporting

Authors have well addressed my questions. I have no more concerns.

Experimental design

no comment

Validity of the findings

no comment

Reviewer 2 ·

Basic reporting

All changes made by the authors are satisfactory.

Experimental design

The authors have made changes that significantly improve the experimental design criteria of this study. The addition of a 12dpf wild-type larvae in Fig. 5B adds a critical point of reference and the authors comments in the rebuttal letter about imaging methodologies used appropriately address the discrepancy between their study/results and those of Eames et al. In addition, the new scoring criteria in Fig. 2 are an improvement from the previous submission.

Validity of the findings

1. Justification for use of appropriate statistical tests for data in Fig. 2 has been adequately described.

2. The only change requested that has not been addressed by the authors surrounds the interpretation of RT-PCR data in the context of nonsense-mediated decay (NMD).

The authors provided a description of the efforts to detect Hdac4 in their tissue samples using available antibodies. While these efforts are appreciated, it is unclear why the authors did not pursue quantitative real-time pcr of cDNA samples, which may have been more fruitful and less work than attempting the antibody approach. Similar band intensities resultant from RT-PCR amplification to saturation is not adequate to make an implication about the presence or absence of NMD. (Even if it were, the data in Fig. 1 is from adult fin clips, which may or may not reflect NMD occurring in larval skeletal tissues.) In my experience, RNAs subjected to up to 90% NMD will still amplify relatively "normally" during RT-PCR, especially if transcripts are abundant in the tissues processed, and the degree to which a transcript may be subjected to NMD may also be tissue dependent.

For these reasons, in addition to the lack of protein data or quantitative RNA analysis, removal of the suggestion that the aik2 and aik3 alleles are not subjected to NMD (or genetic compensation) in the Discussion secton (Page 21, L824-828 of tracked-changes word doc) is warranted. Further, the authors seemingly contradict this implication later in the discussion (Page 21, L842-846 of tracked-changes word doc) by referring to the "haploinsufficient" phenotype (which often results from NMD). The mutant allele may be non-functional, hypomorphic, or even produce a dominant-negative effect manifested by aberrant protein products. The authors have no data to distinguish between these outcomes, and this should be acknowledged more openly in the discussion. This is particularly true given the slightly different phenotypes manifested by the aik2 and aik3 alleles.

·

Basic reporting

While the manuscript is well written overall, I am still not certain that appropriate, consistent gene/gene product nomenclature is used throughout. I defer to the editor on this matter but, for example, line 101 “… cytoplasm, MEF2C becomes unbound to Hdac4, and can activate…” I’m not clear why MEF2C is all caps while Hdac4 only has the first letter capitalized when talking about these two gene products in the same context. I strongly suggest further careful editing to ensure consistent capitalization, and italics are used according to the nomenclature guidelines in zfin/MGI etc.

Similarly, when referring to the zebrafish hdac4 alleles the authors should only say the allele designation once (hdac4^aik3), i.e. reserve the hdac4^aik3/aik3 designation for cases where you are specifically referring to homozygous mutants. This confusing usage is throughout the manuscript. Lastly, use the complete allele name when referring to an allele hdac4^aik3 not just aik3.

The authors should carefully edit to find all cases with the above mis usages and strive for consistency and adherence to convention.

Lines 561-4 “the proportion of approximately one quarter of maternal-zygotic mutants with defects suggests there may effects of other unknown alleles influencing the phenotypic outcome of loss or reductions of hdac4 on neural crest development.” It is confusing to use the word alleles here. To me this implies that there are hdac4 alleles in the background that modify the phenotype. A better word might be unknown loci, or modifiers, or genes.

The text edits in this revision significantly help clarity and readability.

Experimental design

The authors have taken appropriate measures to clearly describe how the different phenotypes are scored. Importantly, given the subtlety of the phenotypes, the blind scoring strategy as described is a crucial improvement.

Validity of the findings

The precocious ossification phenotype is subtle, making it difficult to measure. That said, the improved, blind, scoring strategy and statistics help in this regard and the increased ossification of the ch appears to be valid by these measurements.

The inclusion of Fig. 5B a wild type at 12 dpf is included to make the claim that the MZ mutants resemble the 12 dpf wild types. This image begs the question, what do the MZ or zygotic mutants look like at 12 dpf? Is the precocious ossification still detectable? Quantifying the phenotype at the 12 dpf stage would strengthen the overall claim that ossification is sped up in hdac4 mutants.

The in situ results (Figure 4 A-H, increased runx2 expression) remain particularly subtle and would be greatly strengthened by qPCR. That said, the authors have gleaned all they can from these in situ findings and according to their scoring approach reach valid conclusions.

Additional comments

It is interesting that the authors do not recover homozygous hdac4^aik2 adults. The two alleles are quite similar so I’m not sure why one would be lethal and the other viable. Assuming the backgrounds are identical, this is curious and if the authors have any speculation as to differences between these alleles it might be useful to include in the discussion.

This nice manuscript is an important contribution, especially as an addendum to the MO phenotype that might mislead researchers as to the role of hdac4 in craniofacial development. I look forward to a careful characterization of the severe loss of craniofacial structure phenotype in future studies. It is striking and I agree that the similarities to the fscn1a MZ phenotype are considerable, perhaps offering a lead-in to some novel genetic interactions controlling craniofacial development.

---

## Round 0.3 · accepted · Accept

Thank you for your consideration of the additional reviewer comments and your revised submission. I am happy to now accept your manuscript for publication in PeerJ.

I have only one suggested edit to be added to your final submitted text. On line 115 I believe that if “osteonectin” is referring to the zebrafish gene it should be italicized, even when using the full name.

You will be given the option to make the reviews of your manuscript available to readers. Please consider doing so as this review record can be a great resource for readers of your paper and contributes to more transparent science.

Thank you again for your contribution.

#